# Crystallography and Microstructure of 7M Martensite in Ni-Mn-Ga Thin Films Epitaxially Grown on (1 1 $\bar{2}$ 0)-Oriented Al_2_O_3_ Substrate

**DOI:** 10.3390/ma15051916

**Published:** 2022-03-04

**Authors:** Bo Yang, Zongbin Li, Haile Yan, Yudong Zhang, Claude Esling, Xiang Zhao, Liang Zuo

**Affiliations:** 1Key Laboratory for Anisotropy and Texture of Materials (Ministry of Education), School of Material Science and Engineering, Northeastern University, Shenyang 110819, China; yangb@atm.neu.edu.cn (B.Y.); lizb@atm.neu.edu.cn (Z.L.); yanhaile@atm.neu.edu.cn (H.Y.); zhaox@mail.neu.edu.cn (X.Z.); 2Laboratoire d’Étude des Microstructures et de Mécanique des Matériaux (LEM3), CNRS UMR 7239, Université de Lorraine, CEDEX, 57045 Metz, France; yudong.zhang@univ-lorraine.fr (Y.Z.); claude.esling@univ-lorraine.fr (C.E.); 3Laboratory of Excellence on Design of Alloy Metals for Low-mAss Structures (DAMAS), Université de Lorraine, CEDEX, 57045 Metz, France

**Keywords:** Ni-Mn-Ga thin films, magnetically-induced reorientation, ferromagnetic shape memory alloys, crystallography

## Abstract

Epitaxial Ni-Mn-Ga thin films have been extensively investigated, due to their potential applications in magnetic micro-electro-mechanical systems. It has been proposed that the martensitic phase in the <1 1 0>_A_-oriented film is much more stable than that in the <1 0 0>_A_-oriented film. Nevertheless, the magnetic properties, microstructural features, and crystal structures of martensite in such films have not been fully revealed. In this work, the <1 1 0>_A_-oriented Ni_51.0_Mn_27.5_Ga_21.5_ films with different thicknesses were prepared by epitaxially growing on Al_2_O_3_(1 1 $\bar{2}$ 0) substrate by magnetron sputtering. The characterization by X-ray diffraction technique and transmission electron microscopy revealed that all the Ni_51.0_Mn_27.5_Ga_21.5_ films are of 7M martensite at the ambient temperature, with their Type-I and Type-II twinning interfaces nearly parallel to the substrate surface.

## 1. Introduction

Ni-Mn-based ferromagnetic shape memory alloys have been considered as the promising materials for magnetic field-driven sensors and actuators, new type solid state refrigeration, and thermomagnetic generators [1,2,3,4,5,6,7,8,9,10,11,12]. Under the external magnetic-, thermal-, and force-field stimulus, they usually occur martensite variant reorientation and/or martensitic transformation [9,10,11,12], thus exhibiting multi-functionalities such as giant magnetic field-induced strain [1,2,3,4,5], excellent magnetocaloric effects [6,7,8], and colossal elastocaloric effects, etc. Among these functionalities, the giant magnetic field-induced strain has gained much attention, since both the single crystalline and polycrystalline bulk materials have demonstrated giant magnetic field-induced strain as much as 6–12% and 1% [1,2,3,4,5], respectively. Such giant magnetic field-induced strains also promote the research interest in epitaxial Ni-Mn-Ga thin films, as the films possess a cost-effective advantage in the application of magnetic micro-electro-mechanical systems [13,14,15,16,17,18,19,20,21,22,23]. So far, the magnetically induced reorientation of martensite variant has been reported in the <1 0 0>_A_-oriented Ni-Mn-Ga thin films epitaxially grown on the MgO(1 0 0) substrate and NaCl (1 0 0) substrate. Nevertheless, magnetic field-induced strains have not been achieved yet, since these kinds of thin films are of single crystalline state in the Austenite at evaluated temperatures and transform to self-accommodated martensite at ambient temperature. The self-accommodated martensite microstructure always contains two kinds of distinct microstructures: Type-X and Type-Y, which is composed of several different oriented martensite variants [17,24,25].

Recently, G. Jakob and his coworkers [26,27] proposed that the <1 1 0>_A_-oriented Ni_2_MnGa thin films can be achieved by epitaxially growing on the (1 1 $\bar{2}$ 0) sapphire substrate (Al_2_O_3_). In addition, the martensitic phase in the <1 1 0>_A_-oriented film on Al_2_O_3_ substrate is much more stable than those in the <1 0 0>_A_-oriented film on MgO (1 0 0) substrate. However, the martensitic transformation temperatures of the <1 1 0>-oriented Ni_2_MnGa thin films are between 260 K and 280 K, suggesting that the Ni_2_MnGa thin films are of Austenite at ambient temperature. The magnetic properties, microstructure, and crystal structure of martensite in <1 1 0>_A_-oriented Ni_2_MnGa thin films have not been revealed yet. A. Sharma and his coworkers [28] also prepared Ni-Mn-Ga thin films on (1 1 $\bar{2}$ 0) sapphire substrate (Al_2_O_3_), which is of martensite at ambient temperature. However, the twining interfaces of martensite are not parallel to the substrate surface [29], since the film thickness is of several micron meter.

In the present work, we successfully prepared the <1 1 0>_A_-oriented Ni_51.0_Mn_27.5_Ga_21.5_ films with different thicknesses from 200 nm to 600 nm on Al_2_O_3_ (1 1 $\bar{2}$ 0) substrates by magnetron sputtering. The characterization by X-ray diffraction technique, transmission electron microscopy revealed that all the Ni_51.0_Mn_27.5_Ga_21.5_ films are of 7M martensite at the ambient temperature, and with their Type-I and Type-II twinning interfaces nearly parallel to the substrate surface.

## 2. Materials and Methods

### 2.1. Thin Films Preparation

Ni_51.0_Mn_27.5_Ga_21.5_ thin films with different thicknesses from 200 nm to 600 nm were grown on Al_2_O_3_ (1 1 $\bar{2}$ 0) mono-crystalline substrates by DC magnetron sputtering and using polycrystalline Ni_48_Mn_30_Ga_22_ as the target materials. The target is of 50.8 mm in diameter and 1.5 mm in thickness. Before deposition, the base pressure of the sputtering equipment was vacuumed to below 9.0 × 10^−5^ Pa. In order to obtain high-quality films, the substrate was heated to 650 °C. The sputtering process was conducted under a constant Ar working pressure of 0.15 Pa with an applied power of 70 W. Under this condition, the sputtering rate is of 0.1 nm/s roughly estimated through film thicknesses divided by deposition time. Before the actual depositions, a pre-sputtering was performed for 15 min.

### 2.2. Structure and Microstructure Characterization

The crystal structure and macroscopic crystallographic features were analyzed by temperature-dependent X-ray diffractometer using Cu-*K*_α_ radiation (λ = 0.15406 nm) (Rigaku Smartlab 9 kW, Tokyo, Japan) and a four-circle X-ray diffractometer (Rigaku Smartlab 3 kW, Tokyo, Japan), respectively. A stylus profiler (Veeco DEKTAK 150, Plainview, NY, USA) was employed to measure the film thickness. The microstructures and chemical composition were examined by scanning electron microscopy (SEM, JEOL-JSM 7001F, Tokyo, Japan) and energy dispersive spectrometry (EDS, Bruker XFlash 4010, Berlin, Germany), respectively. The cross-sectional microstructures at nano and atomic scale were characterized by transmission electron microscopy (JEOL JEM 2100F, Tokyo, Japan) working at 200 kV. The cross-sectional sample for TEM characterization was prepared using the focused-ion beam (FIB, FEI Helios nanolab, Hillsboro, OR, USA) lift-out technique. Temperature-dependent magnetization curves and magnetic hysteresis loops were measured by Versalab (Quantum Design, San Diego, CA, USA).

## 3. Results and Discussion

### 3.1. Crystal Structure and Microstructure

Figure 1 displays the conventional and temperature-dependent X-ray diffraction patterns of the Ni_51.0_Mn_27.5_Ga_21.5_ thin films with different thicknesses from 200 nm to 600 nm grown on Al_2_O_3_ (1 1 $\bar{2}$ 0) mono-crystalline substrates. As shown in Figure 1a, it is seen that only one specific diffraction peak of Ni_51.0_Mn_27.5_Ga_21.5_ thin films is observed in the patterns, except the peaks from the Al_2_O_3_ substrate, indicating that the films possess a strong preferred orientation. However, since the planar distance of (2 2 0)_A_ plane of Austenite and (1 $\bar{2}$ $\bar{10}$)_7M_ plane of 7M martensite are roughly equal, we cannot distinguish which specific diffraction peak belongs to the 7M martensite or the Austenite. We performed the temperature-dependent X-ray diffraction measurement, as shown in Figure 1b–d. As can be seen from Figure 1b–d, for all the films, with the increase in temperature, the specific diffraction peak of (1 $\bar{2}$ $\bar{10}$)_7M_ gradually shifts to the low angle side. When the temperature is higher than 350 K, there is a sharp shift to the low angle side in the XRD patterns, which indicated that the (1 $\bar{2}$ $\bar{10}$)_7M_ of 7M martensite transformed to the (2 2 0)_A_ of Austenite. It should be noted that the coexistence of Austenite and martensite phases in the 600 nm-thick Ni_51.0_Mn_27.5_Ga_21.5_ film, which may be due to the restriction from the substrate and the temperature is not high or low enough for the full martensitic transformation.

In order to determine the lattice parameters of the martensite, a four-circle X-ray diffractometer was employed to measure more XRD patterns at various azimuth angles (Phi) and tilt angles (Psi), as shown in Figure 2. With the employment of a four-circle XRD diffractometer, the crystal structure of the Ni_51.0_Mn_27.5_Ga_21.5_ thin films is identified as 7M modulated martensite and of monoclinic. The lattice parameters are shown in Table 1.

Figure 3a–c shows the microstructure of Ni_51.0_Mn_27.5_Ga_21.5_ thin films with different thicknesses taken by FE-SEM. From these FE-SEM images, no twining microstructure of the martensite variants can be observed in the top-view SEM images. All the films can be identified as continuously grown and the surface of the film becomes coarse with increasing thickness of the films. It is worth noting that white areas on the surface of the films can be identified as precipitates and the pit on the films might be defect pores among the column grains.

To further analyze the microstructure and the martensitic configuration of the Ni_51.0_Mn_27.5_Ga_21.5_ films on the Al_2_O_3_ substrate, the cross-section TEM characterization was conducted. As displayed in Figure 4a–c, the martensite is visible in all films. The twin interfaces of martensite are parallel to the substrate surface. A selected area electron diffraction confirmed that the martensite is of 7M martensite (insets in Figure 4a). In addition, with the increase in film thickness, the number of both martensite variants and the twin interfaces increased, but the thickness of each martensite variant remains unchanged. Even though the microstructure of all the films can be identified as column grains, a detailed observation in Figure 4d shows that the twin can pass the grain boundaries, which indicated that the column grain boundaries are small-angle grain boundaries. From Figure 4d–f, martensite can be classified as the hierarchical structure in which martensite lath incorporates martensitic variants. The further crystallographic analysis demonstrated that each plate corresponds to one crystallographic variant, there are a maximum of four-oriented variants in each group and Type-I twin, Type-II twin, and Compound twin are present in the film, which is marked as the dotted line.

### 3.2. Macroscopic Texture

In order to determine the orientation relationship between the variants and the substrate, X-ray diffraction was used to measure the pole figure of the Ni_51.0_Mn_27.5_Ga_21.5_ thin film with a thickness of 600 nm and the Al_2_O_3_ (1 1 $\bar{2}$ 0) substrate. As shown in Figure 5a,b, two symmetrically distributed spots appear at the {0 0 0 1} pole figures of the Al_2_O_3_ substrate and three spots appear at the {1 1 $\bar{2}$ 0} pole figure of the Al_2_O_3_ substrate. Thus, it can be determined that the edges of the substrate are parallel to Al_2_O_3_ [5 $\bar{5}$ 2] and Al_2_O_3_ [$\bar{1}$ 1 1]. Thus, we choose Al_2_O_3_ [5 $\bar{5}$ 2], Al_2_O_3_ [$\bar{1}$ 1 1] and Al_2_O_3_ [1 1 0] as the coordinate direction of the macroscopic sample coordinate system.

As shown in Figure 5c, four spots appear at the {1 $\bar{2}$ $\bar{10}$}_7M_ pole figures of the Ni_51.0_Mn_27.5_Ga_21.5_ thin film. The measured pole figure from the image of Figure 5c is in agreement with the theoretically predicted pole figure from the image of Figure 5d. Since the (1 $\bar{2}$ $\bar{10}$)_7M_ plane of 7M martensite is parallel to the (2 2 0)_A_ plane of Austenite, we can conclude that the films were epitaxially grown on the Al_2_O_3_ (1 1 $\bar{2}$ 0) substrate, which justified that there is one crystallographic orientation for the Austenite in the present thin films. The crystallographic orientation of the Austenite can be described using Euler angle (60°, 90°, 45°). The mismatch relationship and the orientation of Austenite can also be determined, as shown in Figure 6. The Al_2_O_3_ [5 $\bar{5}$ 2] and Al_2_O_3_ [$\bar{1}$ 1 1] direction parallel to the [1 $\bar{1}$ 1]_A_ and [1 $\bar{1}$ $\bar{2}$]_A_ direction of the Austenite of Ni-Mn-Ga thin films.

### 3.3. Magnetic Properties

Magnetization measurements are employed to identify the magnetic field-induced variant reorientation of the twin related martensitic variants in Ni-Mn-Ga thin films. Before the measurement of M-H loops, temperature-dependent magnetization cures were carried out to determine the martensitic transformation temperature as shown in Figure 7a. The M-T curves confirmed that the martensitic transformation temperature of the Ni_51.0_Mn_27.5_Ga_21.5_ thin films are above the ambient temperature, which suggested that these thin films are of martensite state at ambient temperature.

The magnetization hysteresis (M-H) loop of Ni-Mn-Ga thin films was measured at different temperatures, as shown in Figure 7b–d. “Magnetization jumps” in magnetization hysteresis loops can be a sign of the magnetic field-induced reorientation of martensitic variants. However, no obvious “magnetization jumps” can be found in magnetization hysteresis loops of the present Ni_51.0_Mn_27.5_Ga_21.5_ thin films with different thicknesses, from Figure 7b–d, which may be attributed to the high stress from the substrate.

## 4. Conclusions

In summary, we deposited <1 1 0>_A_-oriented Ni_51.0_Mn_27.5_Ga_21.5_ thin films from 200 nm to 600 nm which are of monoclinic 7M at room temperature on the Al_2_O_3_ (1 1 $\bar{2}$ 0). X-ray diffraction and transmission electron microscopy characterization revealed that the martensite-related twin interfaces are parallel to the substrate, and each group possesses a maximum of four-oriented variants. Only one crystallographic orientation for the Austenite is in the thin film. The characterization of magnetic properties shows that the magnetic field-induced variant reorientation was not obvious in the present thin films.

## Figures and Tables

**Figure 1 materials-15-01916-f001:**
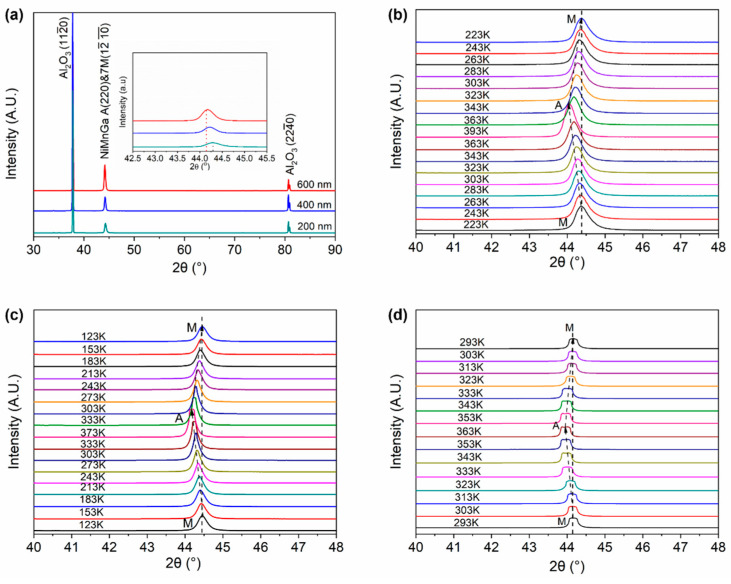
(**a**) X-ray diffraction pattern of Ni_51.0_Mn_27.5_Ga_21.5_ thin films with different thicknesses and (**b**–**d**) the corresponding temperature-dependent XRD patterns. (**b**) 200 nm, (**c**) 400 nm, and (**d**) 600 nm. In (**b**–**d**), A and M represent the Austenite and Martensite, respectively.

**Figure 2 materials-15-01916-f002:**
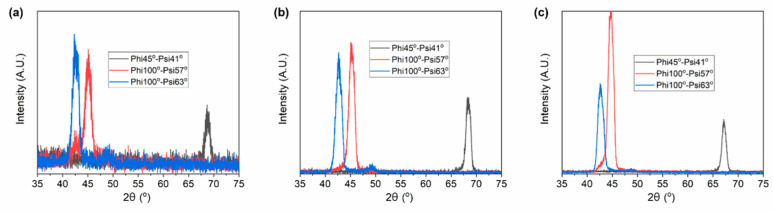
X-ray diffraction patterns of Ni_51.0_Mn_27.5_Ga_21.5_ thin films at various azimuth angles (Phi) and tilt angles (Psi). (**a**) 200 nm, (**b**) 400 nm, and (**c**) 600 nm.

**Figure 3 materials-15-01916-f003:**
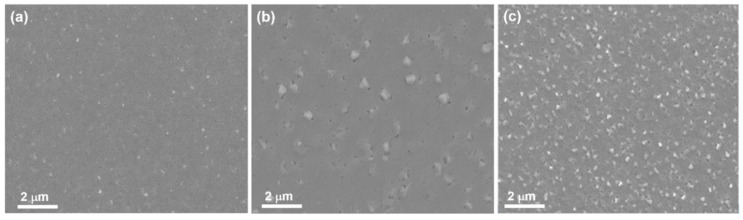
SEM images of Ni_51.0_Mn_27.5_Ga_21.5_ thin films. (**a**) 200 nm, (**b**) 400 nm, and (**c**) 600 nm.

**Figure 4 materials-15-01916-f004:**
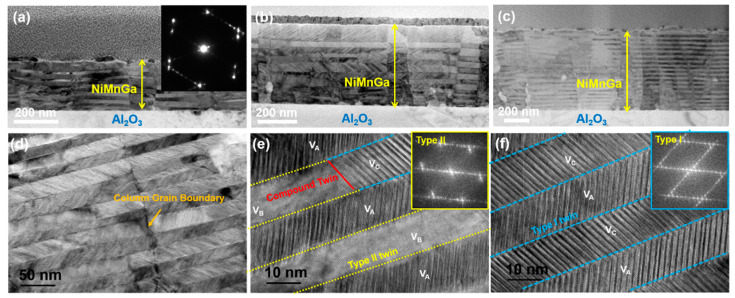
Cross-sectional TEM bright-field images of Ni_51.0_Mn_27.5_Ga_21.5_ thin films. (**a**) 200 nm, (**b**) 400 nm, and (**c**) 600 nm. (**d**–**f**) Detailed analysis of the Ni_51.0_Mn_27.5_Ga_21.5_ thin films with a thickness of 600 nm.

**Figure 5 materials-15-01916-f005:**
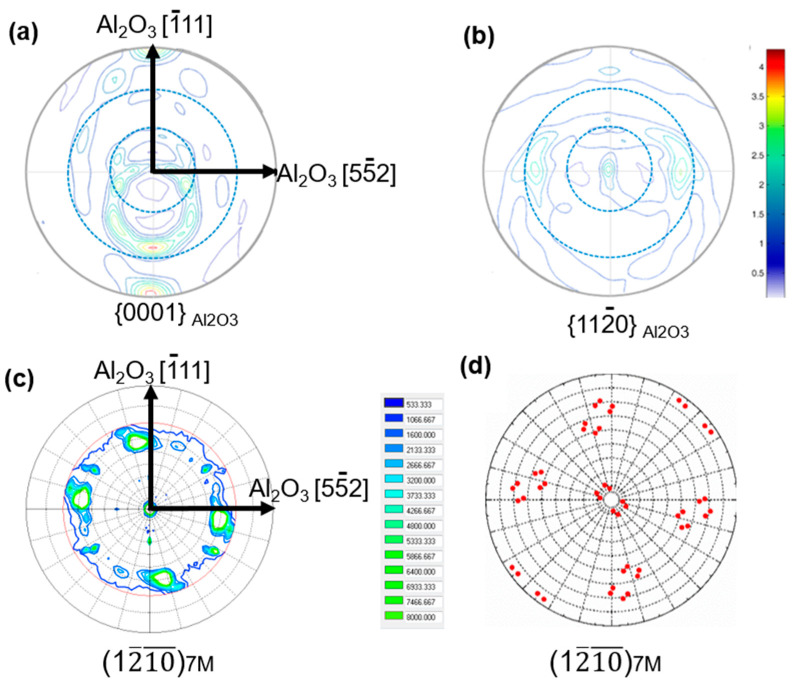
(**a**) {0001}_Al2O3_ and (**b**) {11$\bar{2}$0}_Al2O3_ pole figures of the Al_2_O_3_ substrate, (**c**) measured and (**d**) theoretically calculated {1 $\bar{2}$ $\bar{10}$ }_7M_ pole figures of the Ni_51.0_Mn_27.5_Ga_21.5_ thin films with thicknesses of 600 nm.

**Figure 6 materials-15-01916-f006:**
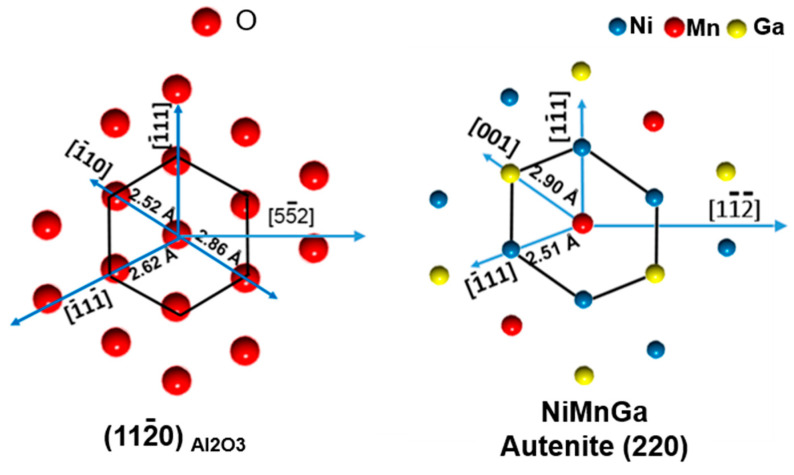
The mismatched relationship between the Al_2_O_3_ substrate and the Austenite of Ni-Mn-Ga.

**Figure 7 materials-15-01916-f007:**
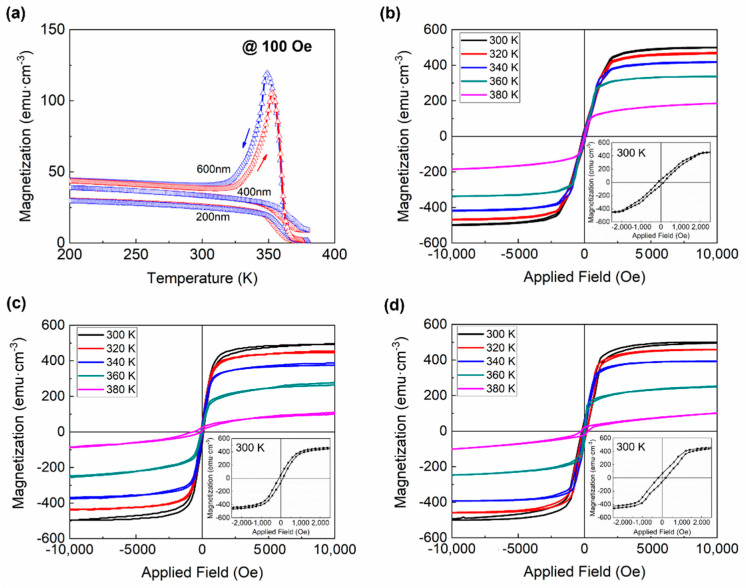
(**a**) Temperature-dependent magnetization cures and (**b**–**d**). Magnetization hysteresis loops at different temperatures of the Ni_51.0_Mn_27.5_Ga_21.5_ thin films. (**b**) 200 nm, (**c**) 400 nm, and (**d**) 600 nm.

**Table 1 materials-15-01916-t001:** The lattice parameters of Ni_51.0_Mn_27.5_Ga_21.5_ thin films on Al_2_O_3_ substrate.

Thickness	Martensite	CS	a (Å)	b (Å)	c (Å)	*β* (°)	V (Å^3^)
200 nm	7M	Monoclinic	4.188	5.446	42.46	93.12	966.9
400 nm	7M	Monoclinic	4.195	5.483	42.34	93.07	972.5
600 nm	7M	Monoclinic	4.164	5.566	42.63	93.03	986.7

## Data Availability

The data presented in this study are available on request from the corresponding author.

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
