# Peer review of "Crystallography and Microstructure of 7M Martensite in Ni-Mn-Ga Thin Films Epitaxially Grown on (1 1 2¯ 0)-Oriented Al2O3 Substrate"

_materials, 2022, doi:10.3390/ma15051916_

Round 1

Reviewer 1 Report

Authors report on the epitaxial growth of 7M martensite Ni-Mn-Ga films on Al2O3(1 1 2Ì… 0) by magnetron sputtering and the characterization of their crystallographic and magnetic properties. The manuscript is worthy to publish in Materials but requires major amendments before acceptance, as outlined below:

  1. Amend the typo ("martenstie") in the paper title
  2. Figure 1d shows a sort of envelop of a double diffraction peak at ca. 44º, pointing to the coexistence of austenite and martensite phases in the 600 nm thick film at all temperatures. Why is it? This should be commented in the paper.
  3. Authors wrote on page 4 "From these FE-SEM images, no martensite twins can be observed in neither the top-view nor cross-sectional SEM images." Remove "nor cross-sectional" because cross-section FESEM images are not shown in the paper or either add "(not shown)".
  4. Specify to which Ni-Mn-Ga film thickness the pole figures shown in Figure 5 belong to in the figure caption.
  5. Please, indicate with an arrow the location of the reversible "magnetization jump" in the hysteresis loops of Figure 7b. More importantly, add a zoom of the loops around the magnetic field this magnetization jump is located and replace the line with symbols because it is not clear to me whether the jump is due to insufficient data points or to a real magnetic field induced reorientation of martensite variants. Most of the times magnetic hysteresis loops show this sort of magnetic jumps because of exceedingly low number of data points. This should be make clear from the plots.

Author Response

Dear Editors and Reviewers,

We are grateful to the reviewer for his/her constructive review on our manuscript (Ref. No. Materials-1559108). According to the reviewer’s suggestions and comments, we have carefully revised the manuscript. For easy localization, the modified parts are highlighted in green in the present version. Hereafter are our responses to the comments of the reviewer.  

Comments 1: Reviewer 1#, Authors report on the epitaxial growth of 7M martensite Ni-Mn-Ga films on Al2O3(1 1  0) by magnetron sputtering and the characterization of their crystallographic and magnetic properties. The manuscript is worthy to publish in Materials but requires major amendments before acceptance, as outlined below: 1. Amend the typo ("martenstie") in the paper title.

Response: Thanks for the reviewer’s comments. We have corrected the typos in the revised manuscript.

Comments 2: Figure 1d shows a sort of envelop of a double diffraction peak at ca. 44º, pointing to the coexistence of austenite and martensite phases in the 600 nm thick film at all temperatures. Why is it? This should be commented in the paper.

Response: Thanks for the reviewer’s comments. We have checked the XRD patterns of the Ni51.0Mn27.5Ga21.5 thin films with thickness of 600 nm. They do show the coexistence of Austenite and martensite phases at all temperatures (as shown in Fig.1S and Fig.1(d), which is due to the restriction from the substrate and the temperature is not high or low enough for the full martensitic transformation. We have added the discussion in the revised manuscript.

Figure 1S. XRD patterns of Ni51.0Mn27.5Ga21.5 thin films with thickness of 600 nm at 293 K and 363 K.

Comments 3: Authors wrote on page 4 "From these FE-SEM images, no martensite twins can be observed in neither the top-view nor cross-sectional SEM images." Remove "nor cross-sectional" because cross-section FESEM images are not shown in the paper or either add "(not shown)".

Response: Thanks for the reviewer’s comments. We have removed “nor cross-sectional” and re-wrote these part in the revised manuscript.

Comments 4: Specify to which Ni-Mn-Ga film thickness the pole figures shown in Figure 5 belong to in the figure caption.

Response: Thanks for the reviewer’s comments. The pole figures shown in Figure 5 belong to the Ni51.0Mn27.5Ga21.5 thin film with a thickness of 600 nm. Indeed, we have measure the pole figures for all the Ni51.0Mn27.5Ga21.5 thin films with different thicknesses (200 nm, 400 nm, 600 nm), they show the similar pole figures as shown Figure 2S. We have added the film thickness in the figure caption.

Figure 2S. The  pole figures of Ni51.0Mn27.5Ga21.5 thin films with different thicknesses (200 nm, 400 nm, 600 nm)

Comments 5: Please, indicate with an arrow the location of the reversible "magnetization jump" in the hysteresis loops of Figure 7b. More importantly, add a zoom of the loops around the magnetic field this magnetization jump is located and replace the line with symbols because it is not clear to me whether the jump is due to insufficient data points or to a real magnetic field induced reorientation of martensite variants. Most of the times magnetic hysteresis loops show this sort of magnetic jumps because of exceedingly low number of data points. This should be make clear from the plots.

Response: Thanks for the reviewer’s comments. We agree with the reviewer’s comments and we have re-measured the magnetic hysteresis loops of the Ni51.0Mn27.5Ga21.5 thin films. The “magnetization jump” in the magnetic hysteresis loop were not obvious with sufficient data points. We have re-wrote the magnetic properties in the revised manuscript.

Reviewer 2 Report

In this work, films with different thicknesses of Ni51Mn27.5Ga21.5 have been grown with preferential orientation (1 1 0) on sapphire (1 1 2Ì… 0) via direct current magnetron sputtering. The films were adequately characterized, revealing twin interfaces parallel to the substrate, and magnetic field induced reorientations were observed on a particular sample.

I would have several questions and suggestions that, if answered, could improve the quality of the manuscript:

Major issues

  1. How was the sputtering rate of 0.1 nm/s estimated? Was there a QCM in the chamber or is it a rough estimation? Some information such as target size and thickness, and applied power seem to be missing from the thin films preparation section which is of absolute necessity for reproducibility issues. Also, is the initial target mono-crystalline or is it poly-crystalline? Was pre-sputtering performed before the actual depositions?
  2. In the Crystal structure and microstructure discussion, in row 114, the authors have identified the crystal structure of Ni50.3Mn28.2Ga21.5 as 7M martensite but in the lattice parameters from Table 1, the composition is Ni51Mn27.5Ga21.5. Why are the compositions different? Do the austenite phase and martensite phase have different compositions? Also, the phrase from 96-99 is not correct on its own (without stating the epitaxial growth relationship) since the deposited film must take the same lattice structure and orientation like that of the substrate (or have defined special epitaxial relations) for it to be epitaxial. From what it seems, it may be preferential oriented growth, not necessarily epitaxy. I would suggest to the authors that in the results and discussion section regarding crystal structure and microstructure they include the spatial groups of their obtained crystals and clearly define the epitaxial growth relationship, to rule out any ambiguity.
  3. Have the authors done an EDS mapping on the surfaces of the films (figure 3) to perhaps observe certain inhomogeneities in the composition of the precipitates compared to the flat surface? The authors should emphasize why these precipitates are being formed and how can their surface density variation be explained with the increasing film thickness.

Minor issues

Row 34, instead of ~occur~ I would suggest ~manifest~

Row 42, instead of ~in the application in~ I would suggest ~in the application of~

Row 78, instead of ~substrate was heat to~ I would suggest ~substrate was heated to~

Rows 104/106, 158/161 there is no need to bar the 0

Austenite is wrote with capital ~A~ sometimes and with small ~a~ othertimes

Author Response

Dear Editors and Reviewers,

We are grateful to the reviewer for his/her constructive review on our manuscript (Ref. No. Materials-1559108). According to the reviewer’s suggestions and comments, we have carefully revised the manuscript. For easy localization, the modified parts are highlighted in green in the present version. Hereafter are our responses to the comments of the reviewer. 

Comments 6: Reviewer 2#, In this work, films with different thicknesses of Ni51Mn27.5Ga21.5 have been grown with preferential orientation (1 1 0) on sapphire (1 1 2Ì… 0) via direct current magnetron sputtering. The films were adequately characterized, revealing twin interfaces parallel to the substrate, and magnetic field induced reorientations were observed on a particular sample.

Response: Thanks for the reviewer’s comments.

Comments 7: I would have several questions and suggestions that, if answered, could improve the quality of the manuscript: Major issues: How was the sputtering rate of 0.1 nm/s estimated? Was there a QCM in the chamber or is it a rough estimation? Some information such as target size and thickness, and applied power seem to be missing from the thin films preparation section which is of absolute necessity for reproducibility issues. Also, is the initial target mono-crystalline or is it poly-crystalline? Was pre-sputtering performed before the actual depositions?

Response: Thanks for the reviewer’s comments. To estimate the sputtering rate, we prepared several Ni-Mn-Ga thin films on the Al2O3 substrate, which are deposited under different sputtering powers (30 W, 50 W, 70 W and 90 W) and for 30 minutes. The thicknesses of these films were measured by a stylus profiler (DEKTAK 150). Then the sputtering rates can be roughly estimated through film thicknesses divided by deposition time. We found that the sputtering rate under 70 W was about 0.1 nm/s. There is no QCM in the chamber in our sputtering equipment. The target is of 50.8 mm in diameter and 1.5mm in thickness. The initial target Ni48Mn30Ga22 alloys were prepared by arc-melting raw Ni, Mn and Ga elements. Before the actual depositions, a pre-sputtering was performed for 15 minutes. We have added the above mentioned information in the revised manuscript.

Comments 8: In the Crystal structure and microstructure discussion, in row 114, the authors have identified the crystal structure of Ni50.3Mn28.2Ga21.5 as 7M martensite but in the lattice parameters from Table 1, the composition is Ni51Mn27.5Ga21.5. Why are the compositions different? Do the austenite phase and martensite phase have different compositions?

Response: Thanks for the reviewer’s comments. There is a mistake with the composition in row 14. All the films are with composition of Ni51Mn27.5Ga21.5. The austenite and martensite have the same compositions. We have corrected these part in the revised manuscript.

Comments 8: Also, the phrase from 96-99 is not correct on its own (without stating the epitaxial growth relationship) since the deposited film must take the same lattice structure and orientation like that of the substrate (or have defined special epitaxial relations) for it to be epitaxial. From what it seems, it may be preferential oriented growth, not necessarily epitaxy. I would suggest to the authors that in the results and discussion section regarding crystal structure and microstructure they include the spatial groups of their obtained crystals and clearly define the epitaxial growth relationship, to rule out any ambiguity.

Response: Thanks for the reviewer’s comments. From the XRD patterns, we can not deduce that the Ni51.0Mn27.5Ga21.5 thin films are epitaxially grown on the Al2O3 substrate. These thin films possess a strong preferred orientation. We have revised it in the revised manuscript.

Comments 9: Have the authors done an EDS mapping on the surfaces of the films (figure 3) to perhaps observe certain inhomogeneities in the composition of the precipitates compared to the flat surface? The authors should emphasize why these precipitates are being formed and how can their surface density variation be explained with the increasing film thickness.

Response: Thanks for the reviewer’s comments. We did the EDS mapping on the surfaces, which is confirmed that the composition is homogeneities, as shown Figure 3S. These precipitates are being formed with increasing of thickness, since the sputtered Ni-Mn-Ga thin films on Al2O3 substrate are of column grains. The increase of film thickness reduced a coarsened surface with high roughness, which is in common for the films prepared by magnetron sputtering.

Figure 3S. EDS mapping of the Ni51.0Mn27.5Ga21.5 thin films with thickness of 600 nm.

Comments 10: Minor issues:

Row 34, instead of ~occur~ I would suggest ~manifest~

Row 42, instead of ~in the application in~ I would suggest ~in the application of~

Row 78, instead of ~substrate was heat to~ I would suggest ~substrate was heated to~ Rows 104/106, 158/161 there is no need to bar the 0

Austenite is wrote with capital ~A~ sometimes and with small ~a~ othertimes.

Response: Thanks for the reviewer’s comments. We have revised these minor issues in the revised manuscript.

Round 2

Reviewer 1 Report

I am fine with the replies given to my queries and the changes introduced in the manuscript. It is a pity that magnetization jumps were finally not present in the M-H loops, but it is better to mention it than publishing something that was incorrect.

Reviewer 2 Report

The authors revised the manuscript accordingly.